# Flavivirus Replication Organelle Biogenesis in the Endoplasmic Reticulum: Comparison with Other Single-Stranded Positive-Sense RNA Viruses

**DOI:** 10.3390/ijms20092336

**Published:** 2019-05-11

**Authors:** Masashi Arakawa, Eiji Morita

**Affiliations:** Department of Biochemistry and Molecular Biology, Faculty of Agriculture and Life Science, Hirosaki University, 3 Bunkyo-cho, Hirosaki-shi, Aomori 036-8561, Japan; iu3219001@hirosaki-u.ac.jp

**Keywords:** flavivirus, endoplasmic reticulum, phosphatidylinositol, ESCRT, reticulons, autophagy

## Abstract

Some single-stranded positive-sense RNA [ssRNA(+)] viruses, including Flavivirus, generate specific organelle-like structures in the host endoplasmic reticulum (ER). These structures are called virus replication organelles and consist of two distinct subdomains, the vesicle packets (VPs) and the convoluted membranes (CMs). The VPs are clusters of small vesicle compartments and are considered to be the site of viral genome replication. The CMs are electron-dense amorphous structures observed in proximity to the VPs, but the exact roles of CMs are mostly unknown. Several recent studies have revealed that flaviviruses recruit several host factors that are usually used for the biogenesis of other conventional organelles and usurp their function to generate virus replication organelles. In the current review, we summarize recent studies focusing on the role of host factors in the formation of virus replication organelles and discuss how these intricate membrane structures are organized.

## 1. Introduction

Many parasitic pathogens invade the inside of host cells in pursuit of an appropriate environment for their growth and replication. The inside of the cell is beneficial for the parasite, not only because of the nutrient-rich condition, but also since it helps the parasite escape the host adaptive immune responses. However, even while inside of the cells, there are several anti-pathogen cellular innate immune responses that may prevent the growth of pathogens. On the other hand, pathogens have evolved to protect themselves from these cellular responses, such as some intracellular pathogens developing the ability to build a kind of nest in order to physically avoid sensors of the host responses, helping to facilitate their efficient growth. For example, some bacteria after invasion convert the membranes of endosomes or lysosomes via its endocytic pathway and then efficiently propagate inside of these compartments [1]. In addition, several viruses are known to induce de novo organelle synthesis in infected cells to promote efficient genome replication and assembly. Since RNA viruses replicate their genome by proceeding through double-stranded RNA intermediates that can present an easy target for host innate immune responses, these organelle formations are critical for the propagation of the viruses [2,3]. 

Single-stranded positive-sense RNA [ssRNA(+)] viruses are known to have properties that deform the endoplasmic reticulum (ER) membrane and create organelle-like compartments called viral replication-organelles [4,5,6,7]. The ER is the largest organelle in the cell and it is a continuous structure; most proteins and lipids that are destined for intercellular compartments are synthesized on the ER membrane [8,9,10]. The ER consists of various structural domains, each of which is associated with a specific function and response to changes in the intracellular environment. The ER is involved in de novo biogenesis of some organelles and ssRNA(+) viruses usurp these functions for their own efficient replication [5,7].

Flaviviruses include mosquito-borne ssRNA(+) viruses, such as dengue virus (DENV), Zika virus (ZIKV), West Nile virus (WNV), Japanese encephalitis virus (JEV), yellow fever virus (YFV), and tick-borne encephalitis virus (TBEV), which cause severe diseases and can potentially lead to death in humans [6,11]. Several diseases caused by flavivirus infections, such as dengue fever or Zika infection, currently pose global public health threats since effective vaccines or drugs to treat these virus infections are still not available [11]. Flavivirus invades host cells via endocytosis and releases its positive-sense RNA genome into the cytoplasm through membrane fusion. Flavivirus encodes a large single polyprotein, which is processed on the ER membrane by the host or viral proteases and is divided into three structural proteins (capsid, prM, and E) and non-structural proteins (NS1–NS5). The capsid protein forms a nucleocapsid complex with the viral genomic RNA, which is important for genome packaging into mature virus particles. The prM and E proteins are significant components of the virus particles and can form spherical virus particles. NS1 has a large ectodomain and is believed to function to deform ER membranes from the luminal side. NS3 is a soluble protein that is anchored to the membrane and has direct interaction with NS2B, which is a transmembrane protein. NS3 has protease activity involved in viral polyprotein processing and RNA helicase activity involved in viral RNA replication. NS5 has RNA dependent RNA polymerase (RdRp) activity and methyltransferase activity and is involved in the replication of the viral genomic RNA and the formation of the 5’-cap structure for protein translation. Viral proteins NS2A, NS4A, and NS4B are membrane-integrated proteins and do not have any known enzymatic activities. The functions of these NS proteins are essentially unknown, but it has been proposed that these proteins are involved in the formation of virus replication organelles [3,6]. After viral genome replication, the newly synthesized viral genomic RNA interacts with the capsid and buds into the ER lumen with prM and E as immature virus particles. These virus particles are then released from the cells via the conventional secretion pathway and undergo the maturation steps in the Golgi and endosomes [6].

In this review, we introduce results from recent studies that focused on ER function and viral replication organelle formation in the cells infected with flaviviruses. In addition, we discuss other ssRNA(+) viruses that are considered to replicate in similar ways to flaviviruses, focusing on the role of host factors in the process of conventional organelle biogenesis.

## 2. Structure of Virus Replication Organelles in Flavivirus-Infected Cells

In flavivirus-infected cells, large compartments containing viral proteins appear in the perinuclear regions (Figure 1). These structures are positive for several ER-membrane markers [12,13] and, therefore, it is believed that they are generated in the ER. Electron microscopy has revealed that these compartments consist of two distinct domains. One of these domains contains an extensive amount of small vesicles, approximately 50–80 nm diameter, with a few dozen of these smaller vesicles being packed in a larger pouch called a vesicle packet (VP) [5,14]. Several electron tomography studies have revealed that the smaller vesicles are not separated from the membrane of the packets [12,13]. These smaller vesicles are simply invaginations, with the inside of the vesicles being connected to the outer side of the packet by a single small pore.

The vesicle packet contains double-stranded RNA (dsRNA), which is an intermediate product of viral genomic RNA replication [14]. Therefore, it appears that the RNA genomes of the viruses are synthesized inside these vesicles. It has been proposed that the viruses create these types of mini-nuclei structures by invaginating the ER membrane and then efficiently replicate their genomic RNA inside of the vesicles. Small pores connecting the vesicle membranes may function in a manner similar to nuclear pores and may be necessary for exporting the newly synthesized genomic RNA to the site of protein synthesis or to the site of particle formation. There is a report, which demonstrates that several components of nuclear pore complexes are detectable in the viral replication organelle of cells infected with hepatitis C virus (HCV), a member of the Flaviviridae family of viruses [15], although for genome replication HCV creates a different type of membrane structure in the ER, which is known as a double-membrane vesicle (DMV). DMVs have an average diameter of 150 nm, and the majority of them are closed structures [16]. Viruses may have developed the ability to create the mini-nuclei structures in order to concentrate materials required for genome replication and to keep dsRNA sequestered from sensor proteins of the host innate immune responses. 

Another domain observed in the viral replication organelle is the convoluted membrane (CM). This domain, which reversibly forms an alternate structure called a paracrystalline structure, appears in infected cells at sites proximal to the vesicle packets [5,12]. Since several membrane-integrated viral proteins accumulate in this compartment, it has been proposed that CMs serve as sites for flavivirus polyprotein translation/processing. However, these structures are induced in a cell-type dependent manner, and their actual function in viral infection remains unclear.

## 3. Phospholipid Subdomains are Required for Viral Replication Organelle Formation

In several genome-wide small interfering RNA (siRNA) screening studies, phosphatidylinositol-4-phosphate (PI-4-P) kinase (PI-4-K) was identified as a host factor required for HCV replication. In HCV infected cells, PI-4-K-III-α interacts with viral NS5A and is recruited to the viral replication site [17,18]. PI-4-K-III-α is the enzyme responsible for catalyzing the phosphorylation of phosphatidylinositol (PI) at the D-4 position to produce PI-4-P. Several lipidome analyses of the lipid composition of HCV replication organelles have revealed that the membranes are highly enriched with PI4P lipids [19]. Significantly, these results were also confirmed in the membranes of replication organelles of poliovirus (PV) [20], a member of the picornaviruses and another ssRNA(+) virus. A yeast strain that is deficient for PI-4-P synthesis shows a defect in the early secretory pathway at the level of the Golgi apparatus [21,22]. To date, many PI-4-P effectors have been discovered with all of them appearing to function on Golgi membranes; either in vesicle budding [23,24] or maintenance of the Golgi structure [25]. Thus, PI-4-P is believed to coordinate protein functions at the Golgi apparatus, which indicates that the protein transport machinery from the ER to the Golgi apparatus may be involved in the formation of virus replication organelles. Currently, three PI-4-P effector downstream proteins have been identified as host factors involved in HCV replication, oxysterol-binding protein (OSBP), ceramide transfer protein (CERT), and Golgi phosphoprotein 3 (GOLPH3) [26,27,28]. Both of OSBP and CERT are known to have a function in lipid transport between the ER and Golgi while GOLPH3 is known to be involved in the maintenance of the Golgi apparatus. These studies suggest that these proteins are involved in the secretion process of virus particles [26,27,28], although the precise mechanisms remain unknown. 

Studies of the enterovirus 71 (EV71), another member of the picornaviruses, have revealed that the recruitment of PI-4-K-III-β to the site of picornavirus-genome replication by viral membrane protein 3A, its binding host factors Arf1 GTPase, and its guanine nucleotide exchange factor GBF1, are essential for the initiation of viral replication organelle biogenesis [29,30]. Results from these studies suggest that small membrane subdomains that usually function in the secretory pathway may be a common platform for replication organelle biogenesis induced by several different ssRNA(+) viruses.

## 4. Factors Involved in the Induction of Membrane Curvature 

Viral proteins in addition to NS5A of HCV and 3A of EV71 have been reported to be involved in the formation of virus replication organelles. The expression of NS4B of HCV or NS2A, NS4A, or NS4B of flaviviruses induces a membranous web, a structure similar to that seen in virus-infected cells [31,32,33]. Therefore, these proteins have been proposed to be the primary factors driving the deformation of ER membranes. However, because these membrane-integrated proteins have no enzymatic activity, their precise functions remain unknown. 

Recently, reticulon (RTN) family proteins were identified as host factors that interact with NS4B of HCV and negatively regulate HCV genome replication [34]. Reticulons are a highly conserved eukaryotic family of proteins that contain two short hairpin transmembrane domains (TMDs) that predominantly occupy the membrane bilayer outer leaflet [35]. Since the short hairpin TMDs of reticulon are able to expand the area of the cytoplasmic outer leaflet relative to the luminal inner leaflet of the bilayer, they generate positive membrane curvature, which is an architectural feature seen in ER tubules [35]. In mammalian cells, the reticulon family of proteins consists of four members, RTN1–RTN4, but the functional differences among the four members are currently poorly understood. It is clear that all the RTN genes are expressed as multiple isoforms, which are generated by alternative splicing [35]. One of the RTN3 splice variants, RTN3A, has been identified as a host factor that interacts to the viral transmembrane protein 2C of EV71 and depletion of RTN3A inhibits EV71 replication [36]. This suggests that the formation of membrane curvatures by RTN3A might be involved in the process of replication organelle biogenesis. 

The RTN family has also been identified by yeast genetics screening as a host factor for ssRNA(+) viruses. Brome mosaic virus (BMV) is a member of the alphavirus-like superfamily and is able to infect the yeast *Saccharomyces cerevisiae.* In infected cells, BMV generates replication organelle-like structures called virus-induced spherules. By screening resistant yeast mutants for viral genome replication, investigators were able to identify RTN homology domain proteins (RHPs) as host factors necessary for spherule formation [37]. Furthermore, Rtn1p, one of the RHPs, interacts with BMV protein 1a [37], suggesting the viruses recruit this protein through direct interaction and that RTN-mediated positive ER membrane curvature formation is required for viral replication organelle biogenesis. 

Similar observations have been reported for the flavivirus WNV. The human RTN3 isoform RTN3.1 interacts with NS4A and depletion of RTN3.1 impairs flavivirus genome replication. Interestingly, the depletion of RTN3.1 induces empty VPs, which are missing the small compartments usually seen inside VPs; this indicates that the generation of small vesicles in VPs is a process dependent on RTN3.1-mediated positive membrane curvature formation [38]. Surprisingly, these results are seen only for WNV and not in DENV-infected cells [38]. In DENV-infected cells, the depletion of RTN3.1 results in the accumulation of elongated long-shaped vesicles in the VPs [38]. These results suggest that RTN proteins are also required for flavivirus VP formation, but this requirement may be dependent on the type of ssRNA(+) virus. 

## 5. Endosomal Sorting Complexes Required for Transport (ESCRT) Pathway and Virus Particle Formation 

In yeast genetics analyses of host factors required for BMV or Tomato bushy stunt virus (TBSV) genome replication, several ESCRT proteins have been identified [39,40]. TBSV is a virus of the Tombusvirus family, which has a positive-sense single-stranded linear RNA genome. This plant virus also infects yeast and has been used as a model system in virology research to study the life cycle of ssRNA(+) viruses. ESCRT proteins were initially identified as factors involved in the transport of the vacuolar proteases [41]. All *S. cerevisiae* mutants evaluated that fail to express any one of the ESCRT proteins exhibit a defect in the formation of intraluminal vesicles (ILVs) of multivesicular bodies (MVBs)/late endosome and also have a defect in the transport of cargo proteins to vacuoles [41]. To date, 17 yeast genes and more than 30 human genes have been identified as ESCRT factors [42]. Each ESCRT factor forms several functionally different subcomplexes, which form an interaction network. This network coordinates the sorting of target cargo proteins and the inward budding of the membrane on MVBs [41]. Cargo proteins that have specific signals (e.g., ubiquitylation) are passed from upstream ESCRT subcomplexes to downstream subcomplexes, which results in the concentration of cargo proteins in a specific area on the endosomal membrane; these process also stimulate the budding of ILVs toward MVB luminal sites [41]. Initially, the ESCRT-0 complex recognizes the cargo signals and recruits its downstream ESCRT-I complex from the cytosol to the endosomal membrane. The ESCRT-I complex then directly interacts with the ESCRT-II complex to transfer the cargo proteins to the downstream ESCRT-III complex. Finally, the ESCRT-III proteins, which include Snf7p, Vacuolar protein sorting 24 p (Vps24p), and Vps2, form circular arrays like a contracting ring on the inside of the budding neck. These events are considered sufficient to induce membrane invagination and ILV formation. Ultimately, the ESCRT complexes are disassembled and disassociate from the membrane via the ATPases associated with diverse cellular activities (AAA) ATPase VPS4, and the internal vesicles are released [41]. This ESCRT pathway is known to function as part of the general membrane fission machinery, including the budding process used by several enveloped virus, the final membrane fission process of cytokinesis, nuclear membrane reassembly, and membrane repair [43]. 

In BMV and TBSV infection, ESCRT plays an essential role in virus genome replication. In cells deficient in ESCRT proteins, BMV and TBSV genome replication are significantly attenuated [39,40]. Furthermore, in Snf7p-deficient cells, spherule formation is also significantly attenuated, suggesting that ESCRT-mediated vesicle formation is involved in the membrane invagination process on the ER during spherule formation [39]. However, for the BMV study, a unique model was proposed in which either viral protein 1a or RTNs remain inside the spherule neck causing the neck to be stabilized and leaving it intact [39]. This is in contrast to the natural role of the ESCRT machinery in which the necks continue to be pinched and separate to form a different membrane [39]. A similar model is suggested in the TBSV study [44]. Currently, the precise mechanistic differences between normal ESCRT function and spherule formation remain unclear. Interestingly, deficiency of ESCRT-0 (Δ*Hse1p*), ESCRT-II (Δ*Vps36p*), an accessory of the pVps4 complex (Δ*Vta1p*), or one of the of ESCRT-III subunits (Δ*Vps60p*) did not significantly affect BMV genome replication [39]. This suggests that some of the specific ESCRT proteins/complexes, but not all, are involved in viral spherule formation. 

Charged multivesicular body protein 4B (CHMP4B), a mammalian ortholog of yeast Snf7p, has been identified using mass-spectrometry analyses as a host factor recruited to JEV and DENV replication organelles [45]. Interestingly, flavivirus genome-replication and VP formation, which are purported to be equivalent to spherule formation in BMV-infected cells, were not significantly affected by the co-depletion of all CHMP4 family proteins [45]. While genome replication was not affected, the levels of infectious virus particle formation were significantly decreased in CHMP4-depleted cells [45]. This suggests that the ESCRT machinery in flavivirus-infected cells is involved in virus particle formation but not VP formation. Similar results are reported in studies of HCV and the picornavirus Hepatitis A virus (HAV) [46,47]. Unfortunately, there is limited information currently available regarding how flavivirus VPs are generated in infected cells. Further experiments are required in order to fully understand these mechanisms.

## 6. Involvement of Autophagosomes and LC3-Positive Compartments 

Autophagosomes are considered ER-derived organelles and are involved in the bulk degradation system in which double-membrane structures sequester cytoplasmic contents for delivery to lysosomes [48]. Several recent studies have revealed that the autophagic pathway is involved in ssRNA(+) virus propagation [49,50,51,52]. Nucleation of PI-3-P-enriched ER subdomains and the subsequent recruitment of a series of Autophagy related gene (ATG) protein complexes are critical for the initiation of autophagosome formation [48,53]. First, the ATG1 complexes and VPS34 complex accumulate for isolation membrane (IM) induction followed by the activation and assembly of two ubiquitin-conjugation systems involving ATG3, ATG5, ATG7, ATG10, ATG12, and ATG16L1 to generate phosphatidylethanolamine (PE) conjugated LC3 family proteins (LC3-II). LC3-II accumulation is involved in IM expansion and closure [48]. 

Autophagy is known to have antiviral activity through the elimination of viral proteins from the cells [54]. However, several studies using picornaviruses concluded the opposite role for autophagy and showed that autophagy promotes the propagation of picornaviruses in cells [49,50,51,52]. PV is a non-enveloped virus, and it is not clear how newly synthesized virus particles are exported out of the infected cells. One of the models proposes that virus particles in the cytoplasm are sequestered by autophagy and then released out of the cells, instead of the natural role of autophagy to deliver the particles to the lysosome for degradation [49]. This model is supported by the following pieces of evidence: 1) Autophagosomes accumulate in the cells in virus-infected cells, 2) viral structural proteins localize with autophagosome, 3) the induction of autophagy enhances virus particle release from the cells, and 4) the inhibition of autophagy impairs virus particle release from the cells [49,50,51,52]. This model is also supported by in vivo experiments using a mouse infection model [55]. Coxsackievirus B3 (CVB3) infection induces autophagosome-like vesicles in pancreatic tissue, and in mice that lack the ATG5 gene, CVB3 propagation in pancreatic acinar cells is significantly decreased and pancreatic pathology greatly diminished. The number of intracellular membrane vesicles was also significantly reduced [55]. 

Another group of investigators proposed a mechanistic model for this pathway [56]. Receptor interacting protein kinase-3 (RIP3), which is an essential kinase for necroptotic cell death signaling, was identified by high-throughput siRNA screening as a positive regulator of CVB3 replication [56]. Depletion of RIP3 diminishes CVB3 propagation by inhibiting autophagosome formation before it is fused to the lysosome. Interestingly, in the later stage of infection, the viral protease 3Cpro targets RIP3 and abrogates RIP3-mediated necrotic signaling [56]. These results suggest that autophagy induction, which could be beneficial for viruses, is a temporal event and that this system is targeted by viral proteases during the later stage of infection in order to prevent cell death. 

Autophagy-mediated virus replication is also proposed for flaviviruses [57,58,59]. Several independent groups have reported that LC3-positive puncta are significantly increased in flavivirus-infected cells [57,58,59] and that these puncta are positive for NS1 protein [60]. Interestingly, the inhibition of autophagy by treatment with 3-methyladenine suppresses flavivirus replication, and induction of autophagy by treatment with rapamycin enhances flavivirus replication [61], suggesting that the autophagy machinery promotes viral propagation in cells. Interestingly, in a DENV study, autophagosome formation was induced during the initial stage of virus infection. However, autophagy flux was blocked by the inhibition of lysosomal fusion during the later stage of infection [62]. This suggests that the requirement of the autophagy machinery can be switched during the progression of viral replication. 

Autophagy is involved in numerous cellular processes [48]. Therefore, we must pay attention to the possibility that the modification of autophagy can affect flavivirus propagation in various ways, the most critical involving the modulation of innate immune responses. Several groups reported that the deficiency of autophagy causes the enhancement of the type-I interferon-mediated anti-viral innate immune signaling pathway [63,64,65]. Another process central to the flavivirus life cycle is lipid metabolism. Flavivirus infection leads to lipophagy or autophagy-mediated lipolysis, which has been thought to promote viral propagation [66,67]. In general, the inhibition of the autophagic pathway attenuates virus propagation. However, it is challenging to interpret the data derived from experiments using autophagy-deficient cells.

LC3 positive compartments that are not autophagosomes have been reported to be involved in the formation of the Coronavirus (CoV) replication organelles [68]. CoVs, such as severe acute respiratory syndrome (SARS) virus and mouse hepatitis virus (MHV), are ssRNA(+) viruses that produce replication organelles called double-membrane vesicles (DMVs), which are believed to be derived from ER membrane-subdomains involved in the secretory pathway [69]. Recently, it was proposed that these DMVs are positive for non-lipidated LC3 and that some proteins are involved in the ER-associated protein degradation (ERAD) pathway [68]. The ER is the site for the folding of membrane and secreted proteins and disturbing this process may cause ER stress and activate a series of signaling pathways collectively known as the unfolded protein response (UPR) [70]. Eventually, unfolded proteins are targeted for degradation by ERAD, which is the system that transports the substrate to the cytoplasm for degradation in the proteasome [71]. ERAD is regulated by several tuner proteins, such as ER degradation-enhancing alpha-mannosidase-like 1 (EDEM1) and Osteosarcoma amplified 9 (OS-9) [72]. Usually, these proteins are constitutively removed from the ER by selective sorting into LC3-I-coated vesicles called ER-derived ERAD tuning vesicles containing EDEM1 and OS-9 (EDEMosomes), which function to deliver their contents to lysosomes for disposal [72,73]. CoV infection induces the accumulation of EDEM1 and OS-9 in LC3-positive DMVs [68]. Removal of LC3, but not the inhibition of the autophagy pathway, prevents CoV propagation, suggesting that CoV hijacks the EDEMosome formation pathway to generate compartments for efficient replication [68,74]. The involvement of the EDEMosome is also suggested for the formation of the flavivirus replication organelle [75]. In JEV-infected cells, virus replication organelles co-localize with endogenous non-lipidated LC3 and EDEM1. Furthermore, depletion of LC3 or EDEM1 by siRNA reduces the level of viral genomic RNA in the cells [75]. The precise mechanisms of EDEMosome formation remain mostly unknown, but elucidation of this pathway may provide information necessary for understanding, not only the mechanisms of replication organelle formation in the ER but also the roles of ERAD and other UPR in replication organelle formation.

## 7. Conclusions and Future Perspectives

The ER is the cellular platform for the biogenesis of various types of organelles. Flavivirus and other ssRNA (+) viruses utilize the flexibility of the ER for their efficient replication by generating specific organelles. The viruses recruit specific host factors that usually have a role in forming specialized subdomains within the ER, such as sites for the formation of the Golgi apparatus, secretory vesicles, ESCRT-mediated invaginations, autophagosomes, and EDEMosomes (summarized in Figure 2). These host factor functions are well conserved and are critical for virus propagation. Therefore, these factors may prove to be strong target candidates for the development of anti-viral drugs.

## Figures and Tables

**Figure 1 ijms-20-02336-f001:**
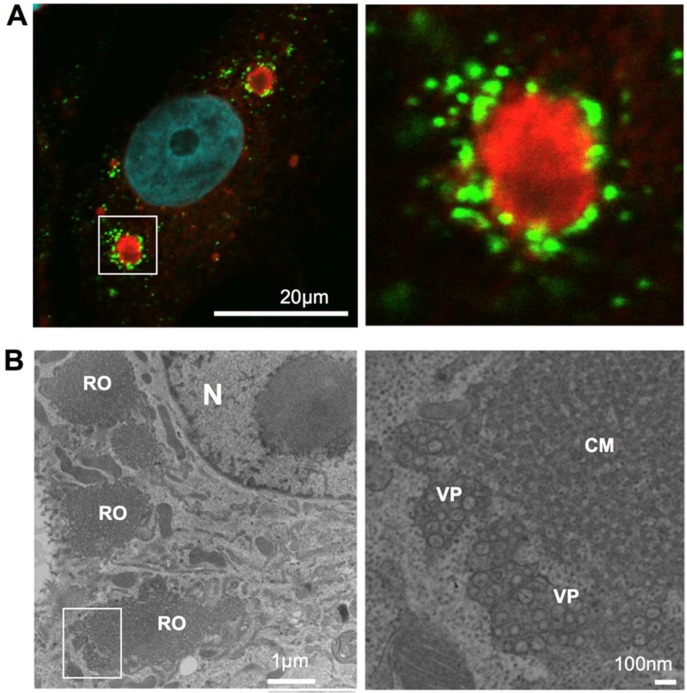
Viral replication organelle in Japanese encephalitis virus (JEV)-infected cells. (**A**). fluorescent imaging of JEV replication organelle. At 48 h post-infection, JEV infected Vero cells were fixed, permeabilized, and stained with anti-nonstructural protein 4B (NS4B; red) and anti-double stranded RNA (dsRNA; Green) antibodies. Nuclear DNA was stained with DAPI (blue). Scale bar: 20 microns. (**B**). Viral replication organelle structure visualized using transmission electron microscopy (TEM). At 48 h post-infection, JEV infected Vero cells were fixed, ultra-thin section, and observed using TEM. N, nucleus; RO, viral replication organelle; CM, convoluted membrane; VP, vesicle packet. Scale bar = 1 micron (left) or 100 nm (right).

**Figure 2 ijms-20-02336-f002:**
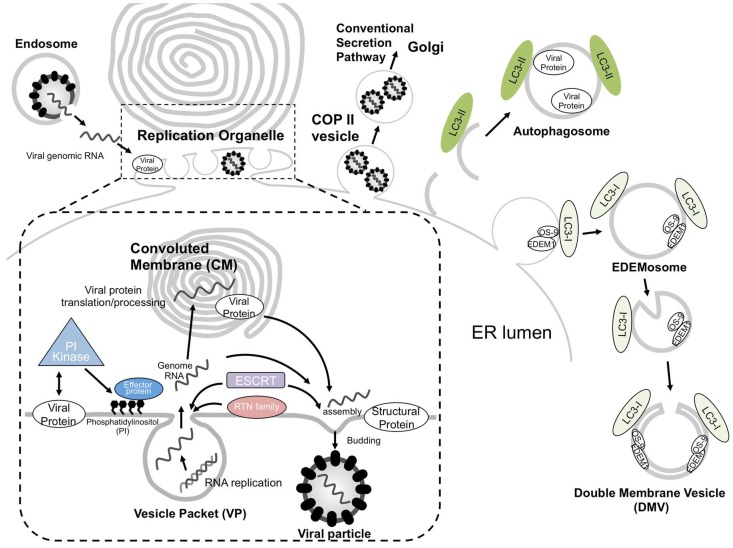
Schematic model of single-stranded positive-sense RNA viruses (ssRNA(+)) virus replication organelle biogenesis in the endoplasmic reticulum. After entry of the virus via the endocytosis pathway, its positive-stranded RNA genome is released into the cytoplasm by membrane fusion. Viral non-structural proteins are directly or indirectly interacted with lipid modifiers, which deform the endoplasmic reticulum (ER) membranes to create vesicle packets (VPs) for the replication of their genomic RNA. Newly synthesized ssRNA(+) is transported outside the VP to be used as a template for further viral protein synthesis or, alternatively, to the site of progeny virus particle assembly. The site of virus particle assembly is generally close to a VP pore. Reticulon (RTN) family proteins are involved in VP formation, and endosomal sorting complex required for transport (ESCRT) proteins can function in VP or virus particle formation. Progeny virus particles are sorted into coat protein complex II (COP II) vesicles and released via the conventional secretion pathway. Apart from this pathway, autophagosomes or EDEMosomes have been proposed to be involved in the formation of the ssRNA(+) virus replication compartment.

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
