# Peer review of "Flavivirus Replication Organelle Biogenesis in the Endoplasmic Reticulum: Comparison with Other Single-Stranded Positive-Sense RNA Viruses"

_ijms, 2019, doi:10.3390/ijms20092336_

Round 1

Reviewer 1 Report

The authors aimed to summarize recent studies describing the involvement of host factors in the biogenesis of replication organelle during flavivirus infection.

1.     Title

A concern of this review is that the title does not match the context in general. While the title indicates a focus on flaviviruses, the authors used a lot of references from studies using other +RNA viruses, including HCV, CoV, and PV. It is therefore advisable for the authors to either change the title to focus more on +RNA viruses in general, or change the context to describe more flaviviruses, in comparison with other +RNA viruses; otherwise the title would appear misleading

2.     Section 1

A brief introduction to the viral life cycle of flavivirus infection is needed to clarify the replication steps that are being referred to in the following context.

3.     Section 2

HCV has a very different replication organelle structure compared to flaviviruses. The purpose of citing only HCV work without alluding to other viruses in this section is unclear.

Lines 104-106: Convoluted membranes and paracrystalline structures are not the same - this line is misleading and should be fixed. Also the authors refer to them as domains - not the correct terminology.

4.     Section 5

The relevance of using a plant virus TBSV as an example needs further explanation. 

5.     Section 6

During the discussion of how autophagy promotes propagation of flaivivirus, a recent work should be cited: Zhang et al.(2018). Flaviviruses exploit the lipid droplet protein AUP1 to trigger lipophagy and drive virus production. Cell Host Microbe 23, 819–831.

6.     Figure 2.

Figure legend is needed to better understand the schematic model. In addition, the viral particle budding step is not precise in this figure. For example, in the case of dengue infection, after replication, progeny virions bud next to the pores of VPs, appearing on membranes that are distinct but tightly apposed to membranes involved in viral replication (Ref 12 in original manuscript). Packed immature virions then accumulate within the lumen of the vesicle packets-containing ER network before transported to Golgi.

Author Response

The authors aimed to summarize recent studies describing the involvement of host factors in the biogenesis of replication organelle during flavivirus infection.

We sincerely appreciate the reviewer’s comments on our manuscript. We have modified the manuscript in agreement with the reviewer’s suggestions. 

1. Title

A concern of this review is that the title does not match the context in general. While the title indicates a focus on flaviviruses, the authors used a lot of references from studies using other +RNA viruses, including HCV, CoV, and PV. It is therefore advisable for the authors to either change the title to focus more on +RNA viruses in general, or change the context to describe more flaviviruses, in comparison with other +RNA viruses; otherwise the title would appear misleading

Response 1: We have modified the title to “Flavivirus replication organelle biogenesis in the endoplasmic reticulum: comparison with other single-stranded positive-sense RNA viruses” in order to make it more representative of the contents of the review. 

2. Section 1

A brief introduction to the viral life cycle of flavivirus infection is needed to clarify the replication steps that are being referred to in the following context.

Response 2: As the reviewer suggested, we added a brief introduction about the flavivirus life cycle, as quoted below:

Line 58-59: Flavivirus invades host cells via endocytosis and releases its positive-sense RNA genome into the cytoplasm through membrane fusion.

Line 73-77:After viral genome replication, the newly synthesized viral genomic RNA interacts with the capsid and buds into the ER lumen with prM and E as immature virus particles. These virus particles are then released from the cells via the conventional secretion pathway and undergo the maturation steps in the Golgi and endosomes[6].

3. Section 2

HCV has a very different replication organelle structure compared to flaviviruses. The purpose of citing only HCV work without alluding to other viruses in this section is unclear.

Response 3: HCV and flavivirus belong to the same family and share many features of their life cycles. Therefore, we considered it appropriate to also cite a study on HCV in this section. However, we understand the reviewer’s concern, and have added the following explanation about HCV double-membrane vesicles (DMVs). 

Line 101-1147:family of viruses [15], although for genome replication HCV creates a different type of membrane structure in the ER, which is known as double-membrane vesicle (DMV). DMVs have an average diameter of 150 nm, and the majority of them are closed structures[16].

Lines 104-106: Convoluted membranes and paracrystalline structures are not the same - this line is misleading and should be fixed. Also the authors refer to them as domains - not the correct terminology.

Response 4: As the reviewer suggested, we replaced this sentence with the following one.

Line 131: This domain, which reversibly forms an alternate structure called a paracrystalline structure, appears …

4. Section 5

The relevance of using a plant virus TBSV as an example needs further explanation. 

Response 5: As the reviewer suggested, we added the following explanations.

Line 211-214: TBSV is a virus of the Tombusvirus family, which has a positive-sense single-stranded linear RNA genome. This plant virus also infects yeast and has been used as a model system in virology research to study the life cycle of ssRNA(+) viruses.

5. Section 6

During the discussion of how autophagy promotes propagation of flaivivirus, a recent work should be cited: Zhang et al.(2018). Flaviviruses exploit the lipid droplet protein AUP1 to trigger lipophagy and drive virus production. Cell Host Microbe 23, 819–831.

Response 6: As the reviewer suggested, we cited the suggested paper and added an explanation about lipophagy as follows:

Line 329-337: Autophagy is involved in numerous cellular processes [48]. Therefore, we must pay attention to the possibility that the modification of autophagy can affect flavivirus propagation in various ways, the most critical involving the modulation of innate immune responses. Several groups reported that the deficiency of autophagy causes the enhancement of the type-I interferon-mediated anti-viral innate immune signaling pathway [63-65]. Another process central to the flavivirus life cycle is lipid metabolism. Flavivirus infection leads to lipophagy or autophagy-mediated lipolysis, which has been thought to promote viral propagation [66,67]. In general, the inhibition of the autophagic pathway attenuates virus propagation. However, it is challenging to interpret the data derived from experiments using autophagy-deficient cells.

6. Figure 2.

Figure legend is needed to better understand the schematic model. In addition, the viral particle budding step is not precise in this figure. For example, in the case of dengue infection, after replication, progeny virions bud next to the pores of VPs, appearing on membranes that are distinct but tightly apposed to membranes involved in viral replication (Ref 12 in original manuscript). Packed immature virions then accumulate within the lumen of the vesicle packets-containing ER network before transported to Golgi.

Response 7: As the reviewer suggested, we added an explanation for the schematic model in the caption of Figure 2. Moreover, we have modified Figure 2 by adding the export pathway of progeny virions.

Line 382-394:Figure 2. Schematic model of single-stranded positive-sense RNA viruses (ssRNA(+)) virus replication organelle biogenesis in the endoplasmic reticulum. After entry of the virus via the endocytosis pathway, its positive-stranded RNA genome is released into the cytoplasm by membrane fusion. Viral non-structural proteins are directly or indirectly conjugated with lipid modifiers, which deform endoplasmic reticulum (ER) membranes to create vesicle packets (VPs) for the replication of their genomic RNA. Newly synthesized ssRNA(+) is transported outside the VP to be used as a template for further viral protein synthesis or, alternatively, to the site of progeny virus particle assembly. The site of virus particle assembly is generally close to a VP pore. Reticulon (RTN) family proteins are involved in VP formation, and endosomal sorting complex required for transport (ESCRT) proteins can function in VP or virus particle formation. Progeny virus particles are sorted into coat protein complex II (COP II) vesicles and released via conventional secretion pathway. Apart from this pathway, autophagosomes or EDEMosomes have been proposed to be involved in the formation of the ssRNA(+) virus replication compartment.

Reviewer 2 Report

Aim of the review was to “summarize recent studies focusing on the role of host factors in the formation of viral replication organelles and discuss how these intricate membrane structures are organized”. In my opinion the purpose of the work has been convincingly achieved, knowledge conveyed in a comprehensible way, complemented with professional photographs and diagrams. The review is very interesting, also from clinical point of view, and may carry practical importance.This review cites all relevant and the latest literature in the field. The very complicated topic is explain very well. All conclusions and future perspectives drawn are comprehensible. Therefore I strongly support publication.I have no substantive (meritoric) critical comments.However, I have only one note about the text. It would be better for the reader to have the list of abbreviations sorted alphabetically.

Author Response

Aim of the review was to “summarize recent studies focusing on the role of host factors in the formation of viral replication organelles and discuss how these intricate membrane structures are organized”. In my opinion the purpose of the work has been convincingly achieved, knowledge conveyed in a comprehensible way, complemented with professional photographs and diagrams. The review is very interesting, also from clinical point of view, and may carry practical importance.This review cites all relevant and the latest literature in the field. The very complicated topic is explain very well. All conclusions and future perspectives drawn are comprehensible. Therefore I strongly support publication.I have no substantive (meritoric) critical comments.However, I have only one note about the text. It would be better for the reader to have the list of abbreviations sorted alphabetically.

Response : We greatly appreciate the reviewer’s favorable view of our work. As the reviewer pointed out, we have arranged the listed abbreviations in alphabetical order.

Round 2

Reviewer 1 Report

The content seems fine; minor textual/grammatical errors exist. It will be useful to have those fixed.